# METCAM/MUC18 Plays a Tumor Suppressor Role in the Development of Nasopharyngeal Carcinoma Type I

**DOI:** 10.3390/ijms232113389

**Published:** 2022-11-02

**Authors:** Yen-Chun Liu, Yu-Jen Chen, Guang-Jer Wu

**Affiliations:** 1Department of Bioscience Technology, Chung Yuan Christian University, Chung Li District, Taoyuan City 32023, Taiwan; 2Department of Radiation Oncology, Mackay Memorial Hospital, Taipei 104, Taiwan; 3Department of Microbiology and Immunology, Emory University School of Medicine, Atlanta, GA 30322, USA

**Keywords:** HuMETCAM/MUC18, NPC-TW01, nasopharyngeal carcinoma type I, migration and invasiveness, 3D basement membrane culture assay, athymic nude mouse model, tumor development, histology and immunohistochemistry, tumor suppression mechanism

## Abstract

From previous studies of negatively correlating the expression of human METCAM/MUC18 with the pathology of nasopharyngeal carcinoma (NPC), we have suggested that human METCAM/MUC18 (huMETCAM/MUC18) might play a tumor suppressor role in the development of nasopharyngeal carcinoma. To scrutinize this hypothesis, we investigated the effects of huMETCAM/MUC18′s over-expression on in vitro cellular behavior and on the in vivo tumorigenesis of one NPC cell line (NPC-TW01). HuMETCAM/MUC18 cDNA was first transfected into the NPC-TW01 cell line, which was established from NPC type I, and many G418-resistant clones were obtained. Then, two NPC-TW01 clones, which expressed high and medium levels of huMETCAM/MUC18, respectively, and one empty vector (control) clone were used to test the effects of huMETCAM/MUC18′s over-expression on in vitro behaviors and on in vivo tumorigenesis (via subcutaneous injection) in athymic nude mice (Balb/cAnN.Cg-*Foxnl^nu^*/Cr1Nar1). The time course of tumor proliferation and the final tumor weights were determined. Tumor sections were used for the histology and immunohistochemistry (IHC) studies. Tumor lysates were used for determining the expression levels of huMETCAM/MUC18 and various downstream key effectors. HuMETCAM/MUC18′s over-expression reduced in vitro motility and invasiveness and altered growth behaviors in 3D basement membrane culture assays, and it decreased the in vivo tumorigenicity of the NPC-TW01 cells. The tumor cells from a high-expressing clone were clustered and confined in small areas, whereas those from a vector control clone were more spread out, suggesting that the tumor cells from the high-expressing clone appeared to stay dormant in micro-clusters. Expression levels of the proliferation index, an index of the metabolic switch to aerobic glycolysis, angiogenesis indexes, and survival pathway indexes were reduced, whereas the pro-apoptosis index increased in the corresponding tumors. The over-expression of huMETCAM/MUC18 in the NPC-TW01 cells decreased the epithelial-to-mesenchymal transition and the in vitro and in vitro tumorigenesis, suggesting that it plays a tumor suppressor role in the development of type I NPC, perhaps by increasing apoptosis and decreasing angiogenesis, proliferation, and the metabolic switch to aerobic glycolysis.

## 1. Introduction

Nasopharyngeal carcinoma (NPC) is a non-lymphomatous, squamous-cell carcinoma, 90% of which develops in the epithelial lining of the posterior nasopharynx cavity [1]. NPC manifests one of three histological subtypes: keratinizing squamous cell carcinomas (WHO type I), nonkeratinizing squamous cell carcinomas (WHO type II), and undifferentiated carcinomas (WHO type III) [1,2]. The early stages of NPC can be effectively treated with radiotherapy [3]. However, most NPC patients are often diagnosed at later stages, for which treatment is much less effective [3]. If these patients are diagnosed earlier or if relapses can be predicted sooner, clinical management would be greatly improved. Thus, there still a great need to search for the specific biomarkers of each type of NPC. One possible candidate is a cell adhesion molecule (CAM), which is aberrantly expressed in NPC since CAMs govern the social behaviors of cells, and the altered expression of CAMs affects the in vitro motility and invasiveness of many tumor cells and in vivo tumorigenesis and metastasis [4]. Until now, examples of this have included the up-regulation of ICAM [5] and the down-regulation of E-cadherin [6,7] and connexin 43 [8] during the progression of NPC. However, these are not type-specific [5,6,7,8], and thus, the search for a type-specific CAM is an on-going endeavor. We found that METCAM/MUC18 may be able to fulfill this need.

Human METCAM/MUC18 (huMETCAM/MUC18), a cell adhesion molecule in the immunoglobulin gene super-family [9,10], is expressed in a normal nasopharynx [11] in addition to several other normal tissues [12,13,14]. HuMETCAM/MUC18 plays an intriguing dual role in the progression of many epithelial tumors [15] as a tumor and metastasis promoter in breast cancer [13,16,17,18] and prostate cancer [19,20,21,22,23,24,25] and as a metastasis promoter in melanoma [26,27,28]; in contrast, it acted as a tumor and metastasis suppressor in ovarian cancer [14,29,30,31] and in one mouse melanoma cell line [32]. However, the role of huMETCAM/MUC18 in the progression of NPC has not been investigated.

For this purpose, we previously initiated the study of whether the altered expression of METCAM/MUC18 correlates with the progression of NPC. We found that huMETCAM/MUC18 was expressed in all of the normal nasopharynx tissues, but it was weakly expressed in 27% of the NPC tissues and not expressed in most of the NPC tissues. Thus, the decreased expression of huMETCAM/MUC18 correlates with the emergence of three subtypes of NPC, suggesting that huMETCAM/MUC18 may function as a tumor suppressor in the development of NPC during the progression of the disease [11]. However, this notion has not been scrutinized both in vitro and in vivo in an animal model.

In this study, we carried out both in vitro tests and an in vivo test in an animal model to vigorously test what role huMETCAM/MUC18 plays in the development of this cancer. We investigated the effect of huMETCAM/MUC18′s over-expression on in vitro cellular behaviors and on the in vivo tumorigenesis of one NPC cell line, NPC-TW01, in athymic nude mice. We found that huMETCAM/MUC18′s over-expression reduced in vitro motility and invasiveness and altered growth behaviors in 3D basement membrane culture assays, and it decreased the in vivo tumorigenicity of NPC-TW01 cells in athymic nude mice, thus providing solid in vitro and in vivo evidence to prove the notion that huMETCAM/MUC18 serves as a tumor suppressor in the development of the NPC-TW01 cell line, which was established from NPC type I. From the preliminary mechanistic studies, we further suggested that the tumor suppression of huMETCAM/MUC18 on NPC cells may be mediated via elevated apoptosis and decreased anti-apoptosis, angiogenesis, proliferation, and the metabolic switch to aerobic glycolysis. In summary, the knowledge learned from this study is useful for understanding the progression of the NPC and for designing therapeutic means to clinically control and treat the cancer [33].

We also carried out parallel studies by using another cell line, NPC-TW04, which was established from NPC type III, and we found that the enforced expression of huMETCAM/MUC18 promoted the development of NPC-TW04 cells, which is opposite to the findings in this report [33,34].

## 2. Results

### 2.1. Expression of huMETCAM/MUC18 Protein in the Clones of the NPC-TW01 Cell Line

As shown in Appendix A, all seven cell lines, including NPC-TW01, expressed a low level of huMETCAM/MUC18 in comparison to that of the SK-Mel-28. For testing the hypothesis that huMETCAM/MUC18 expression might suppress the tumorigenesis of the NPC cells, we used a biochemical method to increase the expression of huMETCAM/MUC18 by transfecting the NPC-TW01 cells with the huMETCAM/MUC18 cDNA via a lipofection reagent, and we isolated the G418-resistant (G418^R^) clones that expressed different levels of the protein. To obtain high-expressing clones, Lipofectamine 2000 (from Invitrogen) was far more superior to three other lipofection reagents (DEMRIE-C (from Life Technology), Lipofectamine (from Invitrogen), and FuGene HD (from Roche)). Figure 1 shows the expression of huMETCAM/MUC18 in three typical G418^R^ clones (#92, #45, and #85) from the high-expressing, intermediate-expressing, and low-expressing groups, respectively, and a vector control clone (V1 clone). The clones #92, #45, and #85 expressed 87%, 47%, and 10% of the protein, respectively, whereas the vector control clone V1 expressed approximately 10 to 20% of the protein (assuming that that of the SK-Mel-28 was 100%).

### 2.2. Over-Expression of huMETCAM/MUC18 Did Not Affect the In Vitro Growth Rate of the NPC-TW01 Cells

We tested whether the over-expression of huMETCAM/MUC18 affected the growth of the NPC-TW01 cells in vitro. As shown in Figure 2, the growth rates of the two clones, clone #92, which highly expressed huMETCAM/MUC18, and the vector control clone, clone V1, were not statistically significantly different. We concluded that the over-expression of huMETCAM/MUC18 in the NPC-TW-01 cells did not alter the in vitro (intrinsic) growth rate.

### 2.3. Overexpression of huMETCAM/MUC18 Decreases the Migration and Invasiveness of the NPC-TW01 Cells

Figure 3a shows that the high huMETCAM/MUC18-expressing clone (#92) had 4.5~5.6-fold lower motility than the vector control clone. Figure 3b shows that the intermediate huMETCAM/MUC18-expressing clone (#45) had 1.75-fold lower motility than the vector control clone and 3.2-fold higher motility than the high huMETCAM/MUC18-expressing clone (#92). Figure 3c shows that the high huMETCAM/MUC18-expressing clone had seven-fold lower invasiveness than the vector control clone. The addition of anti-huMETCAM/MUC18 appeared to increase the motility of the V1 clone, and it increased the invasiveness of all clones. Taken together, we concluded that the over-expression of huMETCAM/MUC18 appeared to inhibit both the in vitro migration and invasiveness of the NPC-TW01 cells, and this was due to the direct effects of METCAM/MUC18.

### 2.4. Over-Expression of huMETCAM/MUC18 Decreased Invasive Growth in the 3D Basement Membrane Assay

We experienced difficulties in demonstrating the formation of an anchorage-independent colony in soft agar (in vitro tumorigenesis) [35] from both clones. Alternatively, a 3D basement membrane assay [36] was useful for testing a similar effect, but it was better for testing the effect on the organization of tumor cells in the 3D culture (Figure 4). Figure 4A shows that clone #92 could form a confined spheroid growth in the 3D basement membrane culture assay. In contrast, the vector control clone V1 formed a more extensive tubular-like growth, as shown in Figure 4C. The confined spheroid growth of clone #92 was changed to the extensive tubular-like growth in the presence of anti-huMETCAM/MUC18 antibody, as shown in Figure 4B. However, the extensive tubular-like growth of the vector control clone V1 was not changed in the presence of anti-huMETCAM/MUC18 antibody, as shown in Figure 4D. Taken together, we concluded that over-expression of huMETCAM/MUC18 appeared to induce a confined spheroid growth of the NPC-TW01 cells in the 3D basement membrane culture assay, which was due to the direct effects of huMETCAM/MUC18.

### 2.5. Overexpression of huMETCAM/MUC18 Decreases Tumorigenesis in Athymic Nude Mice

Previously, the NPC-TW01 cells were shown to yield tumors via subcutaneous (SC) injection of the cells (non-orthotopic injection route) in a Balb/C athymic nude mouse model [37,38]. However, an unusually high number of cells, approximately 10^7^, were used for the injection, which might yield experimental artifacts. To eliminate the possible artifacts due to the injection of an extremely high number of cells, we used a much lower cell number for the injection. By co-injection of the cells with Matrigel, we found that 1.5 × 10^6^ cells, approximately 1/10 of that used previously, were used and were able to induce tumor formation in Balb/C athymic nude mice. Furthermore, we found that an even lower number of cells (1.5 × 10^5^) was also able to induce tumors at the SC site [33]. Tumorigenicity appeared to be similar between the male and female Balb/C nude mice. When 1.5 × 10^6^ cells were used for the injection, tumor formation by the V1 clone was much more efficient than that of clone #92, as shown in Figure 5. Figure 5a shows that tumor formation by the V1 clone was clearly visible at 7 days after injection, more prominent at 28 days, and 16 times larger at 40 days. At the end point (40 days after injection) when the mice were euthanatized, the final mean tumor weight of the V1 clone was 18 times larger than that of clone #92, as shown in Figure 5b,c. When 1.5 × 10^5^ cells were used for the injection, tumor formation by the V1 clone was also much more efficient and visible than that of clone #92, which could not induce any visible tumor in the nude mice [33]. We also tested the in vivo tumorigenicity of clone #45, which expressed a lower level of huMETCAM/MUC18 than that of clone #92, and we found that the tumorigenicity of clone #45 was better than that of clone #92, but it was lower than the vector control clone V1 [33], indicating that there was a dosage effect of huMETCAM/MUC18 on the tumorigenicity of the NPC-TW01 cells. From these results, we concluded that the over-expression of huMETCAM/MUC18 suppressed tumorigenicity and decreased the final tumor weight of the NPC-TW01 clones/cells in an athymic nude mouse model.

### 2.6. Expression of huMETCAM/MUC18 in Tumor Tissues

Figure 6a shows the electrophoretic mobility of huMETCAM/MUC18 expressed in the tumors was similar to that in the three clones grown in vitro in culture and that in the control cell line, SK-Mel-28. We concluded that the proteins expressed in the tumors were not drastically modified or altered from the in vitro cultured clones/cells.

Figure 6b shows the histology (panels A–D) and immunohistochemistry (panels E–L) of the NPC-TW01 tumors. Tumors from the V1 clone were spread out in larger clusters (panels A,B in Figure 6b). In contrast, tumors from clone #92 appeared in many smaller clusters (panels C–D). HuMETCAM/MUC18 antigens were weakly expressed in the centers of tumor clusters induced from the V1 clone (panel F in Figure 6b). In contrast, huMETCAM/MUC18 antigens were strongly expressed in the centers of tumor clusters induced from clone #92 (panels G–H). As also shown in panels G-H, huMETCAM/MUC18 antigens were predominantly expressed on the cytoplasmic membrane, similar to the positive control tumor section from the prostate cancer cell line LNCaP-induced tumors (panel E in Figure 6b). The results show that the tumors were, indeed, induced from the injected NPC-TW01 clones/cells.

### 2.7. Preliminary Mechanisms of huMETCAM/MUC18-Induced Tumor Suppression in the NPC-TW01 Cells

The mechanism by which huMETCAM/MUC18 expression affects the tumorigenesis of NPC cells has not been studied. By deducing from the knowledge we have gained about the tumorigenesis of other tumor cell lines (such as melanoma and breast, ovary, and prostate cancers) induced by huMETCAM/MUC18, huMETCAM/MUC18 may affect tumorigenesis through cross-talk with many signaling pathways that regulate the proliferation, survival, apoptosis, metabolism, and angiogenesis of tumor cells [14,16,18,23,29]. We therefore predicted that the enforced expression of huMETCAM/MUC18 may affect tumorigenesis by altering the expression of its downstream effectors, such as the indexes of apoptosis/anti-apoptosis, proliferation, survival, aerobic glycolysis, and angiogenesis, in the tumor cells. For this purpose, we determined the expression of the levels of Bcl2, Bax, and PCNA and the ratio of phospho-AKT/AKT, LDH-A, and VEGF in the tumor lysates, as shown in Figure 7A. Figure 7B shows the quantitative results that PCNA and the ratio of phospho-AKT/AKT, LDH-A, and VEGF were decreased in the tumor lysates of clone #92. Bax was slightly increased in the tumor lysates of clone #92, and the ratio of Bcl2/Bax was three times higher in the tumor lysates of the vector control clone #V1 than that of clone #92.

Taken together, we concluded that the enforced expression of huMETCAM/MUC18 suppressed the tumorigenesis of the NPC-TW-01 cells by increasing the apoptosis index (Bax) and by decreasing the anti-apoptosis index (Bcl2), proliferation index (PCNA), signal for the survival pathway (ratio of phospho-AKT/AKT), aerobic glycolysis (LDH-A), and angiogenesis indexes (VEGF), which was confirmed by the results of the vascular density, as shown in Figure 8.

## 3. Discussion

To scrutinize the hypothesis deduced from the results of the negative correlation of the expression of huMETCAM/MUC18 with the three histological types of NPC tissue sections [11,33], we tested the effects of the over-expression of huMETCAM/MUC18 in NPC-TW01 cells on their in vitro behaviors and on in vivo tumorigenesis. We showed that the over-expression of huMETCAM/MUC18 reduced in vitro motility and invasiveness and altered the growth behavior in the 3D basement membrane culture assay. We also showed that the over-expression of huMETCAM/MUC18 drastically suppressed the in vivo tumorigenicity of the NPC-TW01 cell line. The huMETCAM/MUC18 in the tumor lysates was not drastically modified or altered, suggesting that the native (unaltered) form of huMETCAM/MUC18 is responsible for these effects. To understand the possible mechanism, we showed that the expression levels of downstream effectors, such as the anti-apoptosis index, proliferation index, index for a metabolic switch to aerobic glycolysis, angiogenesis indexes, and survival pathway index were reduced, suggesting that huMETCAM/MUC18 may mediate tumor suppression via elevated apoptosis and decreased anti-apoptosis, angiogenesis, proliferation, and metabolic switch to aerobic glycolysis. Taken together, we concluded that the over-expression of huMETCAM/MUC18 in the NPC-TW01 cells suppressed the tumor formation induced by the cells, supporting the notion that huMETCAM/MUC18 plays a tumor suppressor role in the development of NPC-TW01 cells, which were established from type I NPC [37,38]. HuMETCAM/MUC18 appeared to play a tumor suppressor role in the development of this type of NPC by mediating elevated apoptosis and decreased anti-apoptosis, angiogenesis, proliferation, and the metabolic switch to aerobic glycolysis.

However, it is not clear how huMETCAM/MUC18 induces tumor suppression via these processes. It is possible that huMETCAM/MUC18 may work through its partner, which somehow downregulates AKT function and its downstream effect on tumor proliferation, similar to the mechanism suggested by a recent published work that an NPC tumor suppressor gene, PTPRG, manifests its function through its partner, EGFR, to decrease the AKT function, which leads to a downstream effect on tumor proliferation [39]. Regardless of the mechanism, the list of cancers in which METCAM/MUC18 plays a tumor suppressor role has been extended from only ovarian cancer [14,29,30,31], mouse melanoma subline K1735-9 [32], and colorectal cancer [40] to include NPC type I (this study).

From our parallel studies of using another cell line, NPC-TW04, which was established from NPC type III, we found that the enforced expression of huMETCAM/MUC18 promoted the development of NPC-TW04 cells, which is opposite to the finding in this study [33,34]. It is not clear why METCAM/MUC18 plays a completely opposite role in different cell lines established from different NPC types. In our previous work, we came to the conclusion that the dual role of huMETCAM/MUC18 in the progression of several human cancers is generally found in the different cell lines of the same cancer type or different cancer types, suggesting that huMETCAM/MUC18 affects the tumorigenesis and metastasis of epithelial tumor cell lines in a very complex way. We further suggested that the one most likely reason was that the different co-factors present in the different cancer cell lines may modulate the role of huMETCAM/MUC18 in these processes via homophilic interactions with endothelial cells and immune cells and via heterophilic interactions with other cells and also with the extra-cellular matrix in the tumor microenvironment [10,15,41]. However, the identification of the corresponding heterophilic ligands of huMETCAM/MUC18 in these cancer cell lines mandates an immediate endeavor in the near future [41].

One point worth special note is that the tumor cells from a high-expressing clone (#92) appeared to be confined in micro-clusters in the tumors, as shown in the results of the HE and IHC (Figure 6B), whereas the tumors induced from the vector control clone V1 appeared to be serious and spread out, suggesting that the tumors from the high-expressing clone appeared to be dormant. Therefore, METCAM/MUC18 may function similarly to other tumor suppressors in other tumor cells [42]. Since tumor dormancy may be due to intrinsic growth inhibition, angiogenic suppression, and/or immunological suppression [42], how METCAM/MUC18 affects tumor dormancy should be an interesting aspect for future investigation. Based on the following evidence, we believe that METCAM/MUC18 may induce tumor dormancy via three aspects: (a) Our results of preliminary mechanistic studies (Figure 7) support the first aspect that METCAM/MUC18 induces tumor dormancy by the inhibition of intrinsic growth. (b) Our preliminary quantitative results of the vascular density (Figure 8) of the SC tumors supports the second aspect that METCAM/MUC18 induces tumor dormancy by the suppression of angiogenesis; however, a definitive conclusion must await future systematic investigations by immune-staining of blood and lymphatic vessels and vasculogenic mimicries with specific antibodies. (c) The third aspect that METCAM/MUC18 may induce tumor dormancy via immunological suppression act may be supported by the expression of METCAM/MUC18 in a subset of activated T cells [43], B cells [44], and natural killer cells [45]. However, the conclusive proof must await further results obtained by the systematic immune-typing of apoptotic cells and the immune-phenotyping of tumor-infiltrating lymphocytes in the METCAM/MUC18-induced tumors. This athymic nude mouse system should be useful for future investigations of how NK cells play a role in the METCAM/MUC18-mediated suppression of the tumor progression of NPC cells [46].

## 4. Materials and Methods

### 4.1. Cell Lines and Antibodies

NPC-TW01 and nine other NPC-TW cell lines were obtained from Dr. Chin-Tarng Lin, Department of Pathology, National Taiwan University, Taipei, Taiwan [37,38]. The NPC-TW01 cell line was established from NPC type I and thoroughly karyotyped by Dr. Chin-Tarng Lin, [37,38]. The chicken anti-huMETCAM/MUC18 antibody was home-made and recognized the internal epitopes of the protein in the region of aa# 212-374 [20]. The antibody could recognize the epitopes of the huMETCAM/MUC18 protein expressed in human cancer cell lines and in formalin-fixed, paraffinized tissue sections [19,20,21,22,23,24,25,26,27,28,29,30,31,32,33,34]. The antibody also has a high specificity, with a minimal cross-reactivity with mouse METCAM/MUC18 protein [47].

### 4.2. Growth of NPC-TW01 and Other Cell Lines

The human melanoma cell line SK-Mel-28 from ATCC was maintained in Eagle’s MEM supplemented with sodium pyruvate, extra non-essential amino acids and vitamins, and 10% fetal bovine serum (FBS). The NPC-TW01 cell line and nine other NPC-TW cell lines were maintained as described in [11,37,38]. All the G418-resistant (G418R) NPC-TW01 clones were grown in the same medium plus 0.4 mg/mL of G418 (Geneticin, GIBCO/Life Technology). All media were obtained from GIBCO/Life technology and the FBS was obtained from GIBCO/ Life technology or Sigma Chemical Co (Burlington, MA, USA).

### 4.3. Lipofection of NPC-TW01 Cells and Selection for huMETCAM/MUC18-Expressing G418^R^-Clones

Lipofection was carried out with Lipofectamine 2000 (1 mg/mL, Cat #11668-019, lot#1024993, Invitrogen) according to the company-suggested procedure, with minor modifications [21]. A total of 0.4 mg/mL of G418 (active component approximately 75%) was added to the growth medium after transfection. After approximately two weeks, G418-resistant (G418^R^) clones appeared and were transferred and expanded sequentially from 24-well to 6-well culture plates. The cell lysate of each clone was made from each well of 6-well plates by Western blot lysis buffer [48], boiled, and kept frozen at -20 C until use for Western blot analysis. The huMETCAM/MUC18-positive clones were further expanded, processed, and frozen in liquid nitrogen as preserved stocks.

### 4.4. Determination of In Vitro (Intrinsic) Growth Rate of NPC-TW01 Clones

The growth rates of the high-expressing G418R clone and vector-control clone were determined by the direct counting of cell numbers after the treatment of monolayers with trypsin, as previously described in [29]. The averages of the hex-plicate cell numbers from each time point were used for calculation. The relative growth rate was determined between three sets of two time-points.

### 4.5. Cell Motility Assay

A cell motility assay was carried out according to a published method by using the Boyden-type Transwell system (Becton/Dickinson Falcon 35-3503) [49], with minor modifications, as previously described in [20]. In brief, 20 h after seeding to the top wells, cells migrating to the bottom wells were detached with trypsin treatment, concentrated by centrifugation, and counted with a hemocytometer [20]. The experiments were repeated three times.

### 4.6. Cell Invasiveness Assay

A cell invasiveness assay was carried out according to a published method [49], with minor modifications, as previously described in [20]. In brief, 24 h after seeding to the top well, of which the bottom was coated with 50 μg of Matrigel (1 mg/mL diluted from 15.95 mg/mL, growth factor-reduced and phenol-red-free grade, Cultrex Basement Membrane Extract Cat #3433-001-01, Trevigen) before seeding the cells, the cells migrating to the bottom well were processed, concentrated, and counted. The experiments were repeated three times.

### 4.7. Three-Dimensional (3D) Basement Membrane (Matrigel) Culture Assay

The published procedures of a 3D embedded basement membrane culture assay [36] were followed, with slight modifications [13]. A total of 5 × 10^4^ cells in 0.3 mL of growth medium were treated with 45 μg of the anti-huMETCAM/MUC18 antibody or control IgY at room temperature for 30 min and then seeded into a well of 8-chambered RS glass slides (BD bioscience chamber slide system), which was pre-coated with 500 μg of Matrigel (basement membrane extract growth-factor reduced with phenol-red, Cultrex Cat # 3431-005-01, 15.95 mg/mL) for three hours. The growth on in the Matrigel was observed daily and photographed with a SPOT digital camera attached to an inverted Nikon microscope.

### 4.8. Determination of the Tumorigenicity of the NPC-TW01 Clones/Cells in Athymic Nude Mice

The guidelines of the National Institutes of Health, USA for the care and use of laboratory animals were strictly followed for all animal experiments, which were reviewed by the CYCU Experimental Animal Care Committee (approval number 10119) and the Mackay Memorial Hospital Experimental Animal Care and Use Committee (approval number MMH-A-S-101-31). In brief, the 3 Rs (reduce, replace, and refine) guidelines were followed for the choice of animals and for maximizing information and minimizing unnecessary studies. The animal housing facility complied with the international standards and was certified, ventilated with sterile caging, and supplied with sterile food and water. Five 37-day-old female or male athymic nude mice (Balb/cAnN.Cg-*Foxnl^nu^*/Cr1Nar1) from the National Laboratory Animal Center, Taipei, Taiwan, were used for the subcutaneous injection of cells from each clone. Single-cell suspensions from the monolayer cultures of the NPC-TW01 cells of clone #92 (p55), clone #45 (p53), or vector clone V1 (p56) were prepared by treatment with trypsin, washed with PBS, and counted by a hemocytometer. Of the cells, 1.5 × 10^5^ or 1.5 × 10^6^ were re-suspended in 0.05 mL of cold DMEM medium without FBS, mixed with an equal volume of 15.44–15.95 mg/mL of Cultrex [49,50], and used for injection in each mouse. The mice were anesthetized by IP injection with Zoletil 50 (50 mg/ mouse) and SC injected with the cells at the back of the neck region or right leg femur by using a gauge #30G1/2 needle. After injection, the well-being of the mice was checked daily and the sizes of the tumors were measured weekly with a caliper until the endpoint of the experiment. There were no adverse effects on the mice bearing tumors until the endpoint of the experiment. Tumor volumes were calculated by using the ellipsoid formula V = π/6 (d1 × d2)^3/2^ (mm) ^3^ [49]. At the endpoint (40 days), all the mice were euthanatized with 100% carbon dioxide in an inhalation chamber and the tumors from each mouse were excised and weighed and a portion was used for making the cell lysate for Western blot analysis [21]. The rest of the tumors were fixed in 10% formaldehyde (Cat #A3684, 2500, Lot# 2T001138, phosphate-buffered histology grade, AppliChem GmbH, Darmstadt, Germany), paraffinized, and sectioned for histology (Leica Microsystems) and immunohistochemical staining.

### 4.9. Western Blot (Immunoblot) Analysis

The cell extracts from the various cultured cell lines were prepared by directly lysing the monolayer cells with the Western blot lysis buffer, as previously described in [20]. The Western blot lysate from each NPC xenograft tumor tissue was prepared from the homogenate of the tissue, as described in [21]. HuMETCAM/MUC18 protein expression in cellular and tissue extracts was determined by the standard procedure of Western blot analysis by using our chicken anti-huMETCAM/MUC18 IgY [20]. The AP-conjugated rabbit anti-chicken IgY (AP162A, Chemicon) was used as the secondary antibody (1/2000 dilution). The same membrane was subsequently used for the detection of the expression of the house-keeping genes of actin and β-tubulin in the lysates, as previously described in [21]. The primary antibodies and AP-conjugated secondary antibodies to detect the expression levels of Bcl2, Bax, PCNA, VEGF, pan-AKT, phospho-AKT (SER437), and LDH-A were previously described in [13,29]. The substrates BCIP/NBT (S3771, Promega) were used for color development. The image of the huMETCAM/MUC18 band and all other bands on the membranes were scanned with an Epson Photo Scanner model 1260 and its intensity was quantitatively determined by the NIH Image J program, version 1.31.

### 4.10. Histology and Immunohistochemistry

Paraffin-embedded tissue sections (5 μm) were de-paraffinized, rehydrated with graded alcohol and PBS, and used for histological staining and immunohistochemical analyses, as previously described in [19,21]. A tissue section of the subcutaneous tumor derived from the prostate cancer cell line LNCaP-expressing clone, LNS239, was used as a positive external control for the immunohistochemical staining [21,23]. The negative controls had the primary antibody replaced by non-fat milk or the control isotype chicken IgY.

### 4.11. Determination of Vascular Density in Tumor Sections

Vascular density in the sections of the SC tumors after HE staining were counted quantitatively under 12 microscope fields and indicated as the number of vasculatures per field.

### 4.12. Statistical Analysis

One-way ANOVA was used for the analysis of the statistical significance of the data in all figures. Two corresponding sets of data were considered significantly different if the *p*-value was <0.05.

## 5. Conclusions

Taken together, the results have proven that huMETCAM/MUC18 plays a tumor suppressor role in the development of NPC type I cells. The tumor suppression of huMETCAM/MUC18 on the NPC cells was not due to decreasing the in vitro (intrinsic) growth rate of the NPC-TW01 cells, but rather, it might be due to the inhibition of the intrinsic in vivo growth via elevated apoptosis and decreased anti-apoptosis, proliferation, survival pathway, and angiogenesis and the rewiring of the metabolic switch to aerobic glycolysis. We further suggest that the tumor suppression effect of METCAM/MUC18 may trigger NPC type I going into tumor dormancy. The knowledge learned is useful for understanding the progression of the NPC and also for designing therapeutic means for the control and clinical treatment of this cancer. Similar to the use of METCAM/MUC18 for the clinical treatment of ovarian cancer, three general strategies may be useful for designing therapeutic means: (a) the reconstitution of the METCAM/MUC18 gene by gene therapy or activation of the tumor/metastasis suppressor genes by activation of the locus on chromosome 11q23.3, (b) directly administering the recombinant METCAM/MUC18 protein to the patients, and (c) aiming at key members in the downstream pathways that are activated by the loss of the metastasis suppressor function [31].

## Figures and Tables

**Figure 1 ijms-23-13389-f001:**
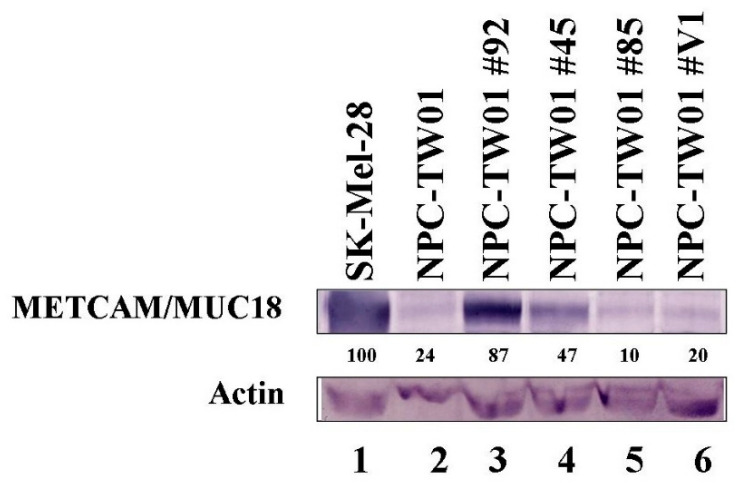
Expression of the huMETCAM/MUC18 protein in the G418^R^-NPC-TW01 clones. HuMETCAM/MUC18 expression in the lysates prepared from the various clones/cells was determined by Western blot analysis, as described in the Section 4. The HuMETCAM/MUC18 expression level in the cell lysate of a human melanoma cell line, SK-Mel-28, was used as a positive control (lane 1) and that from the parental human nasopharyngeal cancer cell line, NPC-TW01, was used as a negative control (lane 2). The HuMETCAM/MUC18 expression in the cell lysates from three huMETCAM/MUC18 cDNA-transfected NPC-TW01 clones (clones #92, #45, and #85) and one vector control clone (#V1) are shown in lanes 3–6. The number under each lane indicates the relative level of huMETCAM/MUC18 in each cell line, assuming that that of the SK-Mel-28 was 100%. Actin is shown as the loading control.

**Figure 2 ijms-23-13389-f002:**
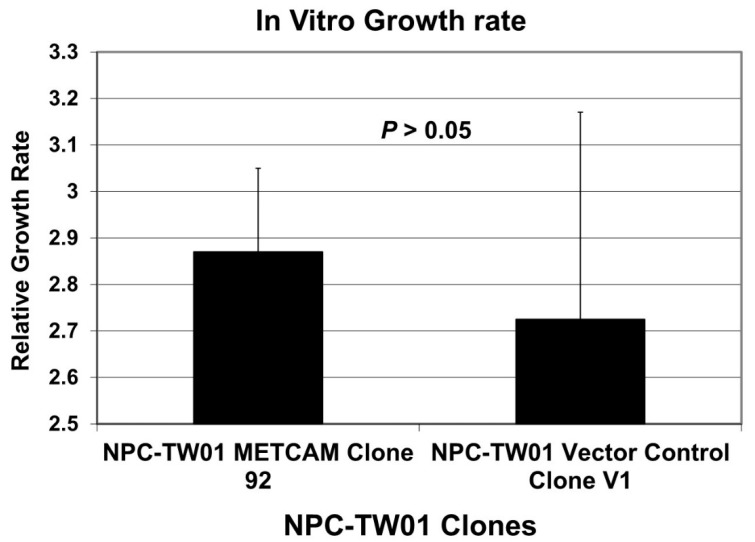
In vitro growth rates of the NPC-TW01 clones. Effects of the huMETCAM/MUC18 expression on the in vitro growth rate of the NPC-TW01 cells were carried out by determining the growth rate of the NPC-TW01 clones 92 and V1 by directly counting the cells at 24, 48, 72, and 96 h after seeding the cells, as described in the Section 4. The mean and standard deviation of the three relative growth rates for each clone were plotted.

**Figure 3 ijms-23-13389-f003:**
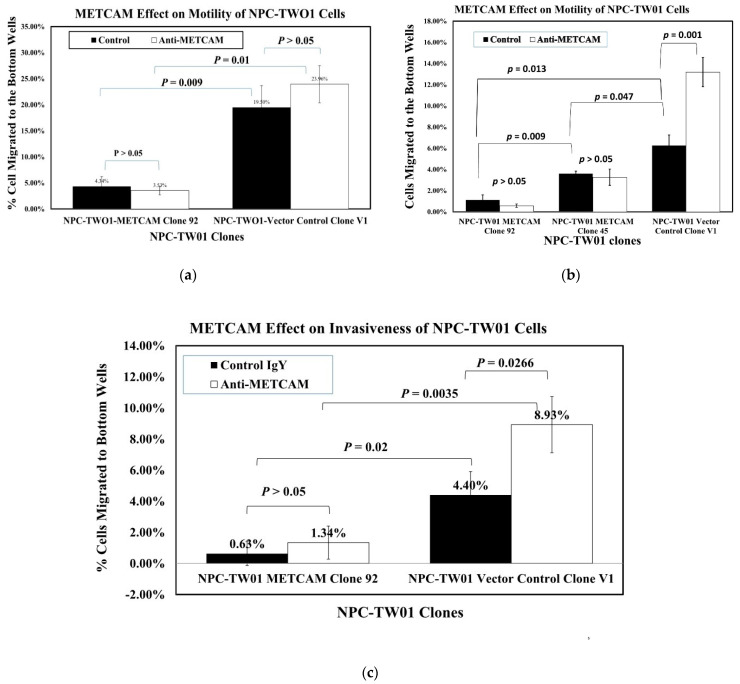
Effects of huMETCAM/MUC18′s over-expression on the in vitro motility (**a**,**b**) and invasiveness (**c**) of the NPC-TW01 cells. All in vitro motility and invasiveness tests were performed in the presence of the anti-huMETCAM/MUC18 antibody or the isotype control IgY, as described in the Section 4 The experiments were repeated three times and the means and standard deviations of triplicate values are indicated. (**a**,**b**) Effects of huMETCAM/MUC18 expression on the in vitro motility of the NPC-TW01 METCAM clone #92 and the vector control clone #V1 (**a**) or the NPC-TW01 METCAM clones #92 and #45 and the vector control clone #V1 (**b**) were determined either in the presence of anti-huMETCAM/MUC18 antibody or the control chicken IgY. The means and standard deviations of triplicate values of the motility tests are indicated. The *p*-values were obtained by comparing the motility of clone #92 with the control V1 clone. The *p*-values were also obtained by comparing the motility of the cells in the presence of the anti-huMETCAM/MUC18 antibody with that of the control IgY. (**c**) Effects of huMETCAM/MUC18 expression on the in vitro invasiveness of the NPC-TW01 METCAM clone #92 and the vector control clone V1 was determined either in the presence of anti-huMETCAM/MUC18 antibody or the control chicken IgY. The means and standard deviations of triplicate values of the invasiveness tests are indicated. The *p*-values were obtained by comparing the invasiveness of clone #92 with the control V1 clone. The *p*-values were also obtained by comparing the invasiveness of cells in the presence of the anti-huMETCAM/MUC18 antibody with that of the control IgY.

**Figure 4 ijms-23-13389-f004:**
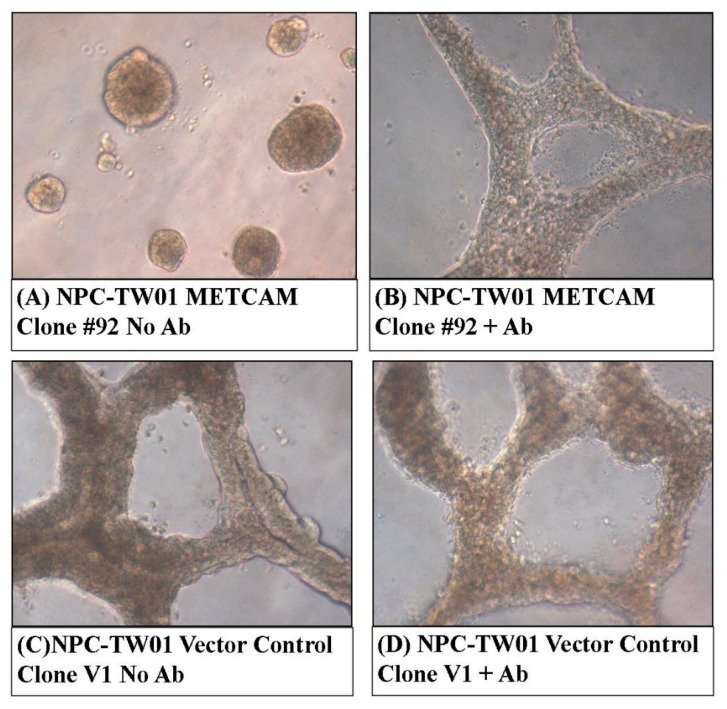
Effects of huMETCAM/MUC18′s over-expression on the growth of the NPC-TW01 cells in the 3D basement membrane culture assay. The growth of the cells from the METCAM clone #92 and the control clone V1 in the 3D basement membrane was observed after 2–9 days and photographed as described in the Section 4. (**A**,**B**) show the growth of METCAM clone #92 in the absence or the presence of the anti-huMETCAM/MUC18 antibody, respectively. (**C**,**D**) show the growth of the vector control clone V1 in the absence or the presence of the anti-huMETCAM/MUC18 antibody, respectively.

**Figure 5 ijms-23-13389-f005:**
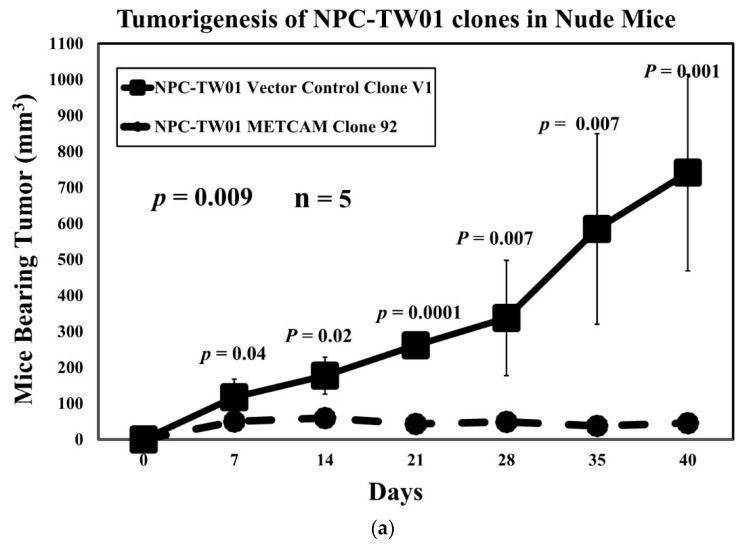
Effects of huMETCAM/MUC18′s over-expression on in vivo tumorigenesis. Tumorigenicity of the METCAM clone #92 and the vector control clone #V1 of the NPC-TW01 cells was determined by subcutaneous injection of the cells from each clone at the dorsal side in male athymic nude mice. (**a**) The tumor proliferation by the two clones by plotting mean tumor volumes/weights versus time after injection is shown. *p*-values were determined between tumor volumes through the time course of the METCAM clone #92 and that of the vector control clone V1. (**b**) The mice bearing tumors of the METCAM clone #92 and the vector control clone V1 and the excised tumors are shown. (**c**) The mean final tumor weights of the two clones in the athymic nude mice were compared at the endpoint. Both the mean final tumor weights from five mice of the vector control clone V1 were statistically significantly heavier than the mean tumor weights from those of the METCAM clone #92 since the *p*-value was 0.007.

**Figure 6 ijms-23-13389-f006:**
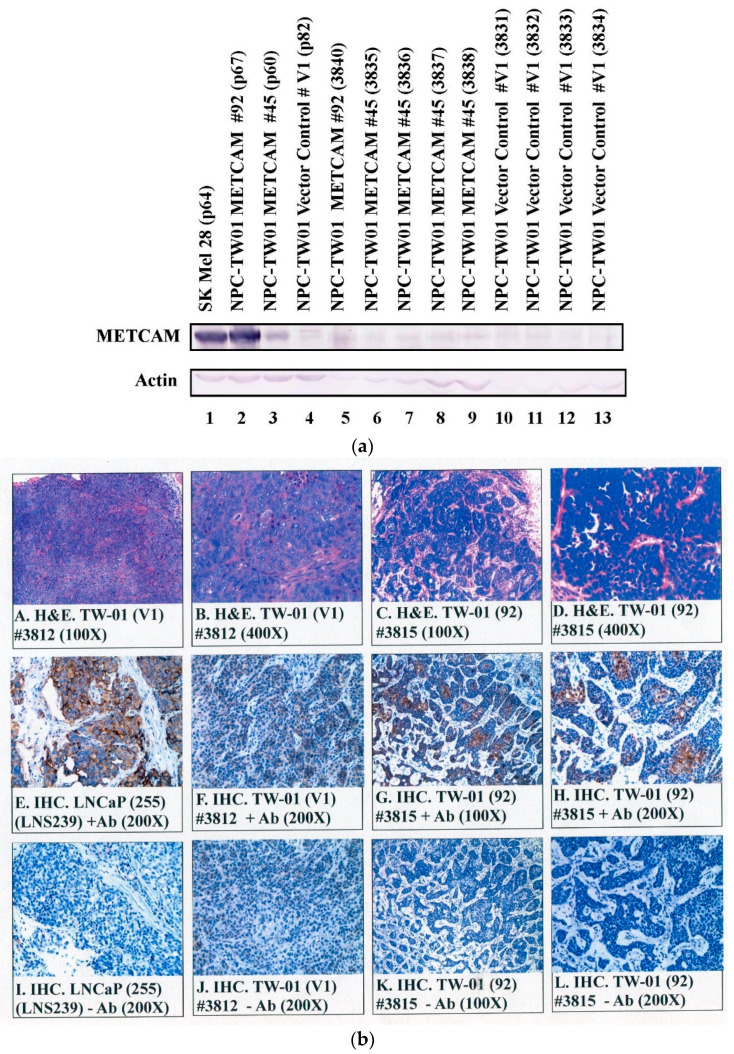
HuMETCAM/MUC18 antigen expression in tumor lysates and in tumor tissue sections. (**a**) The expression of huMETCAM/MUC18 in the lysates from the tumors was determined by Western blot analysis, as described in the Section 4. The expression of huMETCAM/MUC18 in the lysates from the tissue cultured the SK-Mel-28 cells (lane 1) and the NPC-TW01 clones #92 (lane 2), #45 (lane 3), and V1 (lane 4) were used as the controls. The huMETCAM/MUC18 expression levels in the tumor lysates are shown in lanes 5–13. The huMETCAM/MUC18 expression levels in the combined lysate from the tumors of the METCAM clone #92 (lane 5), in the lysates from the tumors of the METCAM clone #45 (lanes 6–9), and in the lysates from the tumors of the vector control clone V1 (lanes 10–13) are shown. As loading controls, the same membranes were reacted with the antibody against the house-keeping gene, actin (as shown). (**b**) The histology and immunohistochemistry (IHC) of the tumors of the NPC-TW01 METCAM clone #92 and the vector control clone V1 are shown. Panels A and B show the histology of the tumor sections from the vector control clone V1 and panels C and D show those from clone #92. Panels E to L show the IHC of these tumor sections. A tissue section of the SC tumors derived from the human prostate cancer line LNCaP-expressing clone (LNS239) was used as a positive external control for the IHC staining (panel E). Panels E to H show the anti-huMETCAM/MUC18 antibody staining of the cells in the tumor sections and panels I to L show the negative controls without the antibody. The tumor section from the METCAM clone #92 showed strong brown color staining in the IHC when the antibody was added (panels G and H); however, the tumor section from the vector control clone V1 showed a weak background staining (panel F). Panels I to L show the corresponding negative controls, which show no staining in the adjacent sections when no antibody was added or when the control chicken IgY was added.

**Figure 7 ijms-23-13389-f007:**
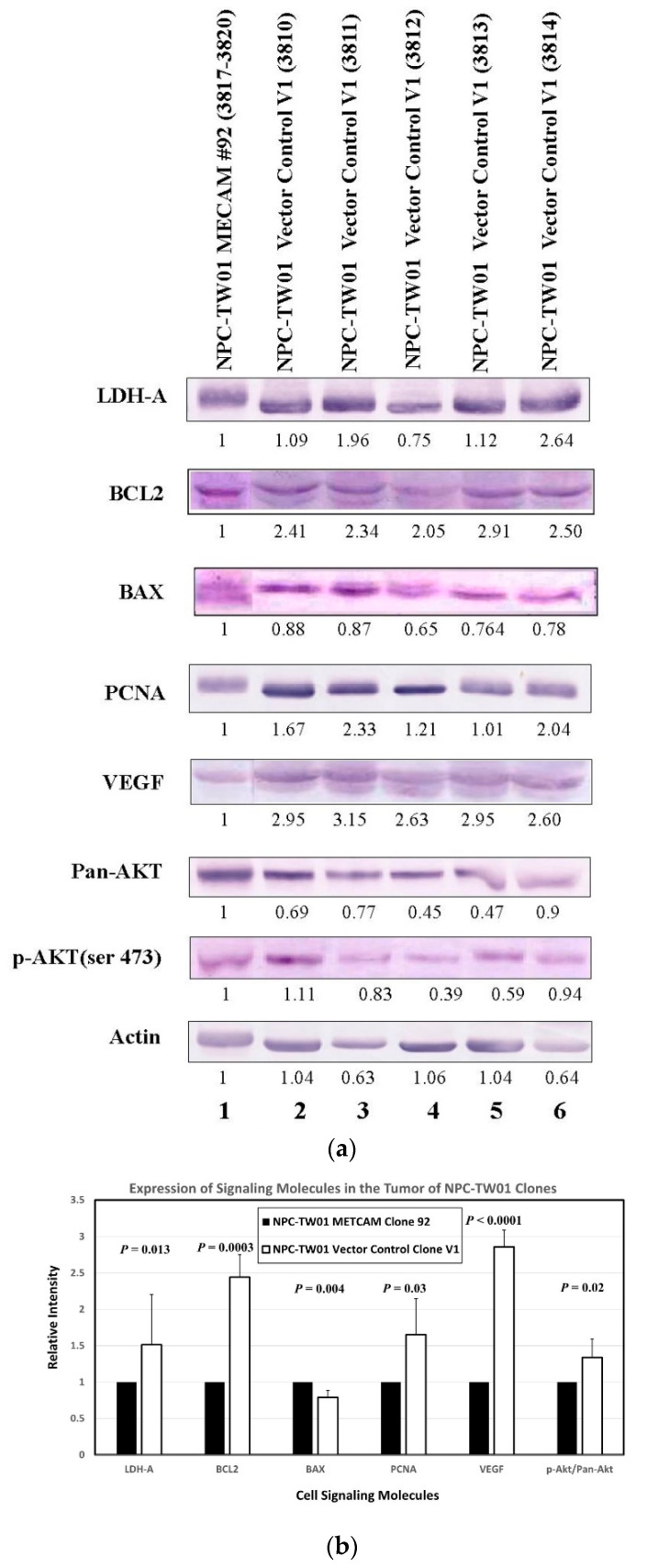
Effects of huMETCAM/MUC18′s over-expression on the expression of key downstream effectors. Tumor lysates were used in the Western blot analysis by using various antibodies, as described in the Section 4. (**a**) The Western blot results of the levels of the various key parameters, such as LDH-A, Bcl2, Bax, PCNA, VEGF, pan-AKT, and phospho-AKT(Ser473), are shown. (**b**) The quantitative results of the above effectors are shown.

**Figure 8 ijms-23-13389-f008:**
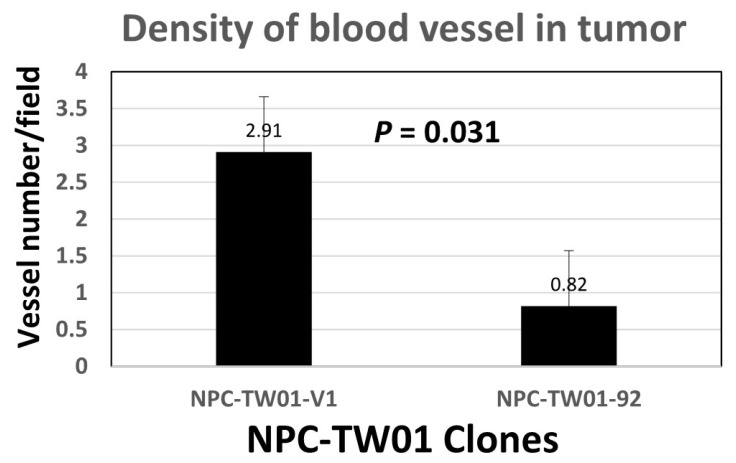
Vascular density in the tumor sections. The vascular density was determined as described in the Section 4.

## Data Availability

The data presented in this study are available on request from the corresponding author.

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
