# Peer review of "METCAM/MUC18 Plays a Tumor Suppressor Role in the Development of Nasopharyngeal Carcinoma Type I"

_ijms, 2022, doi:10.3390/ijms232113389_

Round 1

Reviewer 1 Report

1. The grammar of the text needs to be edited

2. Reorganize according to Instructions for Authors

3. Insufficient resolution of images

4. The content structure needs to be rearranged and adjusted

Reviewer 2 Report

1. The manuscript contains too many sentences or paragraphs copied from the author’s previously published papers (reference 16), including almost the entire Introduction section, the methods section 2.1 and 2.2, etc.

2.The introduction structure needs to be redesigned.

d. The last paragraph of the introduction mentioned that NPC-TW04 was promoted by enforced expression of huMETCAM/MUC18, in complete contrast to NPC-TW01. It is interesting. But what are the mechanisms of huMETCAM/MUC18 expression on NPC cells, what are the differences, and what causes the opposite expression? It could be commented in the discussion section.

4. What are the main differences between this manuscript and reference 16, and what are the new findings of this study?

Round 2

Reviewer 1 Report

The manuscript has been substantially revised. I have no more comments.

Reviewer 2 Report

None.